# Overarm Training Tolerance: A Pilot Study on the Use of Muscle Oxygen Saturation as a Biomarker

**DOI:** 10.3390/s24144710

**Published:** 2024-07-20

**Authors:** Bhargav Gorti, Connor Stephenson, Maia Sethi, KaiLi Gross, Mikaela Ramos, Dhruv Seshadri, Colin K. Drummond

**Affiliations:** 1Department of Biomedical Engineering, Case Western Reserve University, Cleveland, OH 44106, USA; bsg61@case.edu (B.G.); cjs260@case.edu (C.S.); maia.sethi@case.edu (M.S.); klg87@case.edu (K.G.); mar292@case.edu (M.R.); 2Department of Biomedical Engineering, Lehigh University, Bethlehem, PA 18015, USA; dhs223@lehigh.edu

**Keywords:** pilot study, wearables, digital biomarkers, clinical trial, muscle physiology, rehabilitation, overarm movements, training regimen, ulnar collateral ligament

## Abstract

Ulnar collateral ligament (UCL) tears occur due to the prolonged exposure and overworking of joint stresses, resulting in decreased strength in the flexion and extension of the elbow. Current rehabilitation approaches for UCL tears involve subjective assessments (pain scales) and objective measures such as monitoring joint angles and range of motion. The main goal of this study is to find out if using wearable near-infrared spectroscopy technology can help measure digital biomarkers like muscle oxygen levels and heart rate. These measurements could then be applied to athletes who have been injured. Specifically, measuring muscle oxygen levels will help us understand how well the muscles are using oxygen. This can indicate improvements in how the muscles are healing and growing new blood vessels after reconstructive surgery. Previous research studies demonstrated that there remains an unmet clinical need to measure biomarkers to provide continuous, internal data on muscle physiology during the rehabilitation process. This study’s findings can benefit team physicians, sports scientists, athletic trainers, and athletes in the identification of biomarkers to assist in clinical decisions for optimizing training regimens for athletes that perform overarm movements; the research suggests pathways for possible earlier detection, and thus earlier intervention for injury prevention.

## 1. Introduction

The ulnar collateral ligament (UCL), one of the main stabilizers of the elbow joint, provides valgus stability when performing a throwing motion. For baseball pitchers, a UCL injury can occur inside the upper arm (humerus) to the inside of the forearm (ulna) and occurs due to the repetitive stress and torque applied on the elbow ligament. Similarly, softball players, with their comparable overarm throwing motion, volleyball players, with their overarm spikes and serves, swimmers, with their freestyle and butterfly strokes, and football quarterbacks undergo the same repetitive overarm movements that generate significant stress on the UCL [1]. Compared to contact-related injuries, research has shown that a larger proportion of throwing-related injuries resulted in more playing time being lost. Moreover, throwing-related UCL injuries have a higher requirement for surgery compared to contact-related injuries [2]. Thus, athletes who repeatedly execute overarm movements in their sport have a significantly higher proportion of severe injuries compared to contact athletes.

Over the past couple of years, ulnar collateral ligament reconstruction, also known as Tommy John surgery, has become a common procedure in athletes ranging from adolescent to elite-level. The number of UCLR procedures has increased from only 1 surgery performed in 1986 to 31 performed in 2012 in MLB players [3]. The current rehabilitation measures for treating UCL injury involve subjective assessments (pain scales) and objective measures such as monitoring joint angles and range of motion. However, after undergoing UCL reconstruction, baseball pitchers are seen to return to roughly 90% of their previous level of play [4].

Although reconstruction surgery is a major contributing factor to the decreased performance, the entire kinetic chain of throwing and high-effort overhead motions in athletics must be considered. Muscle atrophy, the bodily tolerance rates, and the fatigue limits of each contributing factor of the kinetic chain can be attributed to decreased performance, higher reinjury rates, and chronic pain [5]. During the motion of throwing a baseball, the main contributors to the tissue biomechanics of the arm are the ulnar collateral ligament, anterior and exterior deltoid, rotator cuff, and labrum [1]. To assist in the development of novel strategies to predict the internal arm physiology of athletes, there are ongoing motion and kinetic modeling studies that attempt to derive factors that contribute to an increased risk of injury. For example, there was a recent systematic review that analyzed some psychological factors, such as loss of interest and fear of reinjury, which affected return to play (RTP) protocols after UCL reconstruction for high school baseball players [6].

### 1.1. Mental Resilience

The challenge of re-injury rates, return to play procedures, and effective injury rehabilitation for UCL reconstruction goes beyond the physiological aspects. A big component of the whole injury process is an athlete’s mental resilience. The mental blockades that persist in the recovery process, such as the fear of reinjury and the loss of interest in putting in the necessary work to get better, can influence whether the athlete can or will return to the level at which they competed before. As seen in previous research studies, mental health disorders in athletes are associated with an increased risk of injury with prolonged recovery times, higher rates of injury recurrence, decreased rates of return to sport, and reduced performance upon return [7]. Overcoming mental hurdles is a crucial aspect of successful rehabilitation, and thus the development of a model that could predict training tolerance levels for baseball pitchers during rehabilitation, including both physiological and psychological parameters, would help to provide clinical and biological long-term benefits. This is why the development of a model that could predict training tolerance levels for baseball pitchers during rehabilitation would help to provide clinical and biological long-term benefits.

### 1.2. Pain Tolerance

Understanding injury and its severity in athletes involves not only physical but also psychological and subjective factors. Pain tolerance, coping strategies, performance level, personal motivation, the importance of competition, and athletic identity all contribute to how injuries are defined and perceived [8]. Pain tolerance, in particular, significantly impacts athletic performance. Athletes often confront the critical decision of when pain becomes intolerable, necessitating cessation of competition. The level of pain an individual can endure directly influences their performance. In addition to tolerating the physical pain of an injury, there are multiple psychiatric risks from injury that an athlete may face. How much one identifies as an athlete can play a role in defining an injury. The importance of the competition can sway an athlete’s decision to hide and manage an injury or severe pain on their own due to the intensity of their desire to compete. When one heavily identifies as an athlete, an injury can be perceived as a failure and/or a barrier to competing, which will increase the emotional burden of the injury [9]. There can be a decrease in self-identification with the athlete role and decrease in self-esteem [10].

### 1.3. Blending Subjective and Objective Assessments

Current research advocates a strategy of combining subjective and objective assessments to understand baseball athletes’ performance and injuries [11,12]. Paci et al. utilized both subjective and objective assessments during a Division I collegiate baseball athlete team’s preseason to assess its predictive value for injury development across the season [13]. Here, Paci et al. utilized upper-extremity, functional movement assessments to objectively assess the athlete’s range of motion prior to the season and subjective assessments such as the Kerlan–Jobe Orthopaedic Clinic (KJOC) score for athletes to subjectively report pain levels when throwing. Additionally, Mehta et al. conducted a retrospective study evaluating the relationship between chronic workload, injuries, and subjective arm health evaluation in a sample of high-school baseball athletes [14]. Here, the MotusBASEBALL sensors (Motus Driveline, Massapequa, NY, USA) were used throughout the pre-season and regular season over the course of three years; these were then utilized to calculate chronic workload, and arm health was assessed immediately after each throwing session. The results of the study indicated that higher chronic loads were associated with increased injury risks. Moreover, a significant relationship was found between throwing-related injuries, subjective arm health, and internal load. Thus, while this research presents a valuable insight into how chronic workload risks can provide information on injury risks, the ongoing, continuous monitoring of collegiate baseball athletes during their season to both better understand and predict athlete performance and injury risk at this level is still lacking.

### 1.4. Research Gap

In the past, models like the 1976 Calvert’s Model have been used to attempt to predict competitive performance based on cardiovascular health, strength, skill, and psychological aspects [14]. However, these predictive models, including Calvert’s, have not been validated due to small sample sizes and an incomplete understanding of injury determinants [15]. The development of wearable systems offers a non-invasive ability to monitor athletes in real sport environments, providing real-time feedback [16]. Athletes have shown an interest in wearable sensors for identifying biomechanical fatigue and early intervention to prevent injury during training and competitive matches [17]. Despite their potential, wearable systems have limitations, such as the need to place devices at specific anatomical locations, the frequency of data sampling, the monitoring of only a few selected variables, and uncertainty about the accuracy of data interpretation [18]. Unique to commercially available sensors is the difficult data management issue of extracting data and harmonizing all the data into a single data file [14]. Also, there is little research on biomechanical approaches to monitoring the progress of movement aiming to achieve injury prevention for sports athletes [19]. This very broad issue is intractable without narrowing the scope of the work, so the present work limits our research scope to objective measures and relationships between muscle fatigue and workload as measured by wearable technology.

While the dataset of Vandrico [20] indicates that more than 250 companies offer over 431 commercially available “wearable sensors”, the number of continuous, noninvasive localized measurements sensors relevant to sports research is actually quite limited [21]. In the context of the current study, two promising wearable sensors for measuring upper-extremity throwing motion are the Moxy Monitor (“Moxy”) sensor for measuring muscle oxygen saturation SmO2 (Fortiori Design LLC, www.moxymonitor.com/, accessed on 1 March 2023) and the Driveline Pulse (“Pulse”) throwing workload sensor (www.drivelinebaseball.com, accessed on 1 March 2023). The Moxy sensor measures muscle oxygen levels and total hemoglobin during training using near-infrared spectroscopy (NIRS) through the skin, providing insight into the muscle metabolism. For the current study, the Moxy sensor was chosen for SmO2 measurements based on device validation clinical trials [12,22,23,24]. Reviews of workload sensors for “swing sports” (volleyball, tennis, baseball) are informative about sensor research efforts [11,25,26,27]; however, commercially available systems specific to baseball are limited. Still, a few workload studies pointed to the success of the Pulse sensor in field tests [28,29,30] and thus the Pulse sensor was chosen for the current research effort. The Pulse offers an in-depth analysis of pitching workload and biomechanics metrics, providing insights into throwing intensity, which can be useful during rehabilitation. Heart rate monitors are often used to determine exercise intensity [31] and the outcome of a recent comparison of devices [32] indicated that the Polar 10 system (www.polar.com, accessed on 1 March 2023) would provide the capabilities needed in the present work. The novelty in selecting our sensor suite lies in their proven validation and specificity to baseball. The Moxy sensor provides reliable SmO2 measurements, while the Pulse sensor offers a detailed analysis of pitching workload and biomechanics. The Polar heart rate monitor was chosen for its ability to accurately assess exercise intensity, which is crucial for rehabilitation insights.

In summary, it can be said that there is potential for commercially available wearable sensors to provide data to assess and evaluate preventive measures against UCL injury biomechanical fatigue, thus enabling early intervention. Within that general context, the aim of the present study is to explore how traditional biomechanical workload measurements can be complemented by muscle metabolism measurements (SmO2) as a proxy for fatigue (often a precursor to injury).

The current research seeks to partially address the gap in the existing clinical research described earlier, as our data build on the data available for other research teams to study; this helps to pave the way for future research on training tolerance models for baseball pitchers and other athletes who perform repetitive overarm movements.

## 2. Materials and Methods

### 2.1. Experiment Design

Our experiment is intended to provide a comprehensive view of the participants’ physical performance and subjective experience, helping to determine the efficacy of the training regimen and its potential effects on players’ performance in game-like scenarios. Because our experiment involves several phases of activity, an overview of the experiment is provided in this section for clarity, with more details provided in Section 2.3.

The four overall components of the test procedure are as follows:Throwing phase;Deltoid exercise phase;.Heart rate monitoring;Subjective assessment

The objective of the throwing phase was for participants to perform ten throws at “game” speed, executed with more than 80% effort, reflecting a game-like intensity. The deltoid exercise phase targets the deltoid muscles and involves eccentric movements involving elevated internal rotation, front deltoid raise, and elevated external rotation. Through repetition (without a rest between sets), these exercises are a form of a stress test, leading to strain on the ulnar collateral ligament. Heart rate monitoring, with a target peak above 130 BPM, ensures participants are actively and intensively engaged in the throwing and deltoid exercise phases.

After each exercise, each players’ subjective perception of their exertion was self-assessed using the Borg Rate of Perceived Exertion (RPE) scale. This subjective measure helps correlate players’ physical feelings with their performance data.

In summary, the collected data include:Performance metrics (accuracy, speed, etc.) from the throws based on data from the Driveline sensor.Heart rate data to monitor physical exertion, based on the Polar 10 sensor.Muscle oxygen data on the anterior deltoid muscle based on the Moxy sensorSubjective ratings of perceived exertion.

All data were secured in an HIPAA-compliant survey and identified by the team for analysis. Microsoft Office (2023), MATLAB (2023), and R [33] were used to analyze the data. By comparing these data points, researchers aim to understand how subjective negative assessments (feeling overly exerted or fatigued) impacted performance and physiological responses during the trial.

For the plots of arm speed and arm torque, a simple linear regression was performed using the average values from each player’s three throws. Linear regression was utilized to generate a “trend line”; the slope of this trend line offers insights into the relative change in performance for each player.

### 2.2. Subject Recruitment

The study was conducted in accordance with the CWRU Institutional Review Board guidelines. All participants were informed about the study and its requirements before volunteering. Participation requirements included no history of labrum or UCL injuries, at least six months post-operation, clearance to play by the NCAA and CWRU athletics (with valid physicals and sufficient concussion testing), active participation in baseball or throwing activities within the past two months, and being between 18 and 22 years old. Volunteers were randomly selected at the coach’s discretion and notified upon selection. To account for different playing positions, the subjects included both pitchers and position players, with five pitchers and three position players among the eight participants.

### 2.3. Data Collection

All participants were equipped with the Moxy, Polar, and Driveline Pulse sensors to record the athletes’ muscle oxygen saturation, heart rate, arm speed, and torque, respectively. For each participant, the Moxy sensor was placed on the anterior deltoid muscle, which contributes to producing the flexion of the arm during throwing motions (Figure 1) [5].

The sensor was securely positioned using the silicone holder provided with the Moxy, and then athletic adhesive was layered to decrease shifting during throwing actions. The Pulse sensor was placed two finger lengths down from UCL on the anterior bundle of the arm and the Polar sensor was placed across the sternum (an inch to the left below the pectoral muscle). For recording purposes, the sensors were only activated during the data collection and did not collect data during warmup periods. After the completion of the test, sensors were immediately stopped and removed for data collection.

### 2.4. Exercise Protocol

Since Section 2.1 only provided an overview of the experimental design, a more detailed description of the exercise protocol is necessary. Before starting the experiment, participants were introduced to the test procedures to facilitate a smooth transition to the actual testing. The warm-up routines were not standardized; participants either followed the baseball team’s routine or stopped when they felt adequately prepared to exert the required effort. Additionally, participants were asked to rate their sleep quality, report if they had practiced the previous day, and assess any arm soreness. These measures were taken to evaluate the potential cumulative effects of these factors, particularly if the subjective assessments were negative. Table 1 summarizes the seven exercises involved in each player’s test. The test procedure is as follows:Throwing phase: Participants perform ten throws at “game” speed from the standard professional and collegiate pitching distance of 60 feet 6 inches.Exercise circuit: After the throws, participants complete a circuit of deltoid exercises using either a DriveLine resistance band or a 5 lb weighted **disc**. The exercises are Elevated Internal Rotation, Front Deltoid Raise, and Elevated External Rotation.Repetitions and sets: Each exercise is performed eight times, with no rest between sets. After completing the exercises, participants immediately begin another set of throws. This sequence is repeated two more times for a total of three sets.Heart rate monitoring: Participants’ heart rates are monitored using a Polar sensor to ensure active participation, with a target peak above 130 BPM.Effort level: Throws should be executed with more than 80% effort, reflecting game-like conditions from the participants’ perspectives.

This protocol ensures an assessment of each participant’s performance and physical exertion that is appropriate for a pilot study. Following each exercise in the protocol, participants were asked to rate their perception of exertion, utilizing the Borg Rate of Perceived Exertion (RPE) scale.

## 3. Results

A total of eight healthy collegiate baseball athletes from Case Western Reserve University participated in the study across the 14-week season (Table 2).

### 3.1. Arm Speed

The arm speed values (Rpm) were recorded through the Driveline Pulse Sensor for each subject. In Figure 2, each graph is represented as a Box Whisker plot where each trial, consisting of ten throws each, is shown. The arm speed for each subject represents the measurement of the peak total rotational speed of the forearm when the subject is pitching. For Trial 1, the arm speed values range from 650 Rpm to 950 Rpm. For Trial 2, the arm speed values range from 750 Rpm to 800 Rpm. For Trial 3, the arm speed values range from 700 Rpm to 900 Rpm. The graphs below demonstrate an overall representation of the arm speed values over three trials.

The linear regression of arm speed trends does not follow a consistent pattern, with both positive and negative regression slopes with no evident correlation to player position.

### 3.2. Arm Torque

Similar to the arm speed measurements, the arm torque values (Nm) for each subject were recorded using the Driveline Pulse Sensor. The data are displayed as Box Whisker plots, Figure 3, showing the results of each trial, with each trial consisting of ten throws. Arm torque measures the Valgus stress on the elbow during a pitch. The results, along with the mean and standard deviation, are provided in each plot. The graphs indicate that arm Varus torque increased with pitching effort, suggesting a kinetic parameter that may be related to the risk of UCL (ulnar collateral ligament) injury, and thus a potential biomarker.

Unlike the trend lines for arm speed, the tendency of the means of each trial featured a consistently positive slope, reflecting the increase in arm torque as each of the trials were performed. Table 3 illustrates that the mean of the regression slope, m, for each of the subjects was m = 4.1, although the overall range was quite broad, ranging nearly an order of magnitude from m = 0.3 to m = 9.0. Of general interest was a measure of overall change in torque between the first and last trial, and the overall percentage change in mean value is also shown in Table 3. These results indicate a consistency in arm torque that is absent in the arm speed, and this suggests the two parameters are decoupled in this pilot trial.

### 3.3. Muscle Oxygen Saturation

Muscle oxygen saturation (SmO2) data were recorded over an eight-minute period for each subject, as shown in Figure 4. Although there seems to be a patten in the time series, combining all plots on one graph was not informative due to the variance in data. Thus, given that the trends shown in Figure 4 can be a bit difficult to discern, one time series plot was selected for clearer visualization, as shown in Figure 5, which details the timing of each exercise type (throwing and rotational) for Participant 1.

Correlating throwing and rotational activity with SmO2 in Figure 5 provides an insight into the trends and their relation to activity. The series graphs indicate that the initial SmO2 percentage ranged from 60% to 100% during the first set of ten throws. During the subsequent rotational exercises, the SmO2 percentage dropped by 20% to 50%. When performing the second set of ten throws, the SmO2 percentage increased by 20% to 40%. Following this, the SmO2 levels varied among subjects based on individual muscle oxygen levels. By the end of the trial, the final SmO2 percentages ranged from 0% to 40%. Overall, the average muscle oxygen saturation during the six trials of throwing and rotational exercises was around 50%. These data highlight the dynamic changes in muscle oxygen saturation in response to different types of physical activity and underscore the variability between individuals. The study’s findings provide insight into how muscle oxygen levels fluctuate during repeated sets of exercise and the specific trends associated with different arm motions.

### 3.4. Heart Rate

The graph in Figure 6 displays the heart rate (bpm) of all eight participants over an eight-minute trial period. These data provide a quantitative measure of how each athlete responds to different types of workouts, offering insights into their physical condition. At the start of the trial, participants’ heart rates ranged from 60 to 140 bpm. As the trial progressed, heart rates either increased or decreased depending on the type of exercise performed. Typically, heart rates increased during pitch throws, reflecting an increase in cardiac output. Conversely, heart rates decreased during rotational exercises. Despite these fluctuations, all participants maintained heart rates within a safe range throughout the trial, indicating their overall fitness. These heart rate data highlight the dynamic response of the cardiovascular system to different exercises and underscore the participants’ ability to handle varying physical demands safely. This information is valuable for understanding how different types of workouts affect heart rate and can be used to optimize training regimens to improve athletic performance and ensure safety.

## 4. Discussion

A key objective of this pilot study was to evaluate the effectiveness of wearable near-infrared spectroscopy technology in measuring digital biomarkers like muscle oxygen saturation and heart rate.

The study focused on the intense movements involved in throwing a baseball to identify key biomarkers in the anterior deltoid and ulnar collateral ligament (UCL). The goal was to use the data collected to help injured athletes and those at risk of injury. Using the simulation model developed by Buffi et al. [5], the study evaluated the kinetic chain of the arm during high-intensity throwing motions by measuring biomarkers in the anterior deltoid and UCL. The main findings showed that, based on the data plots, muscle oxygen saturation (SmO2) ranged approximately from 30% to 70% during pitching exercises, compared to approximately 25% to 60% during rotational exercises. Previous research has indicated that intense training and muscle-strength exercises can cause changes in SmO2 levels [14,15]. This study’s results showed increased SmO2 levels throughout all exercise bouts, regardless of type, suggesting that the changes were due to the high-intensity nature and duration of the workload. Although heart rate was measured, it did not show significant clinical relevance beyond aligning with the SmO2 data. However, heart rate was crucial for assessing participant exertion to control effort levels.

Indeed, arm torque and arm speed do correlate with muscle oxygen saturation levels, but only arm torque showed a clear association. The data revealed an important trend: most participants’ muscle oxygen saturation levels correlated with both arm torque and arm speed. This suggests that SmO2 and arm torque could be viable biomarkers for modeling high-intensity arm motions in baseball players.

Arm torque trend was most evident when analyzing each set of throws and workouts for each trial against the overall set of three trials. This is demonstrated by examining Figure 2, Figure 3 and Figure 4, where a higher third-quartile box for every subject suggests that, as trials progress and fatigue accumulates in the anterior deltoid, more torque is measured, leading to increased stress on the ulnar collateral ligament. Previous studies have shown that repetitive strain can lead to ligament overuse and changes in ligament viscoelasticity. Consequently, the ligaments adjust their extracellular matrix component stiffness, affecting the muscle’s ability to generate the necessary torque [17,18]. This repeated stress, especially without proper exercise protocols and adequate recovery, can heighten the risk of ulnar collateral ligament injury [19]. During overarm movements like pitching, the elbow experiences significant load through external valgus torques, compressing the lateral side and tensile forces on the medial side of the elbow. The UCL’s primary function is resisting these external valgus torques. Therefore, our data provide insight into how overarm movements, particularly pitching, impact the UCL. Although no single combination of biomarkers can fully assess UCL health, monitoring SmO2 alongside arm torque and speed may offer valuable insights into the strain placed on the ulnar collateral ligament during high-intensity movements.

According to the NCAA, pitchers are the most commonly injured players on the field [34]. Specifically, the shoulder and the elbow joints are the most commonly injured joints for softball pitchers. Similar to baseball throwing, increasing repetitive throws causes more fatigue, which results in a higher risk of injury or reinjury.

### Future Work: Softball

The mechanics of softball pitching are different from overarm throwing and baseball pitching. Softball pitches are commonly described as an underhand or “windmill” pitch. The arm goes through different movements, such as an arm circle, a cocking phase where the arm is bent at the elbow with the wrist cocked back, an acceleration phase where the arm extends and the wrist snaps to generate speed on the ball, and a release phase where the ball is then released and spun from the fingertips. The arm makes a complete circular motion, with the peak shoulder extension occurring when the throwing arm is somewhat past the top of the circular motion. In general, the glenohumeral joint rotates the arm through an axis that is perpendicular to the humerus diaphysis [34]. In the early stages of the pitch, there are low-magnitude kinetic and kinematic forces [34]. When the pitcher starts the windup and strides forward, there is higher activity in the supraspinatus, infraspinatus, deltoid muscle, and teres minor [34]. This is where the humerus is internally rotated, since it is flexed at the shoulder. The acceleration phase includes less supraspinatus activity, but the anterior deltoid, infraspinatus, and teres minor are active. The highest magnitude kinetic and kinematic forces are before the release phase when the arm is moving down, snapping the ball (delivery phase). The pectoralis major, subscapularis, and biceps brachii are most active and the humerus is flexed and internally rotated.

The different pitching motions can result in different injury risks. Even though the throwing mechanics are different, the forces on the arm are similar [35]. The peak forces in baseball occur in the deceleration phase, whereas, in softball, the peak forces occur in the delivery phase [26]. Both mechanics place torque on the elbow, where stress is placed on the medial elbow and UCL. Forces are slightly higher with the overhead throwing mechanism, but more torque is generated with the windmill pitch [35]. In one study, it was found that the forces were similar even though there were differences in the injury patterns of elbows in softball and baseball pitching [34]. However, the forces are smaller in softball due to the shoulder position relieving some of the elbow stress. In the release phase, the elbow moves from an extension position to a flexion, whereas in baseball, the overhand position causes the elbow to flex and then extend. Additionally, there is less abduction and external rotation in the shoulder compared to baseball mechanics [34]. Since less torque is experienced by UCL and more torque is experienced on the shoulder, further analysis of biomarkers surrounding the shoulder joint may prove beneficial. However, overall research revolving around the UCL can be accredited to a compounding factor.

## 5. Conclusions

The findings from this study support the ongoing use of wearable technology for assessing player performance and monitoring the health of the ulnar collateral ligament (UCL) during high-intensity movements. While individual wearable data collection is valuable, combining data from multiple sensors to derive a single biomarker remains challenging. The preliminary data gathered in this study offered insight into potential biomarker combinations for clinically monitoring the health and performance of baseball players across different age groups, although further data collection is necessary. Continued research is crucial, as future studies could build upon our preliminary findings to investigate UCL injuries and performance tolerance more comprehensively.

The trends observed in the data collected during this pilot study underscore the importance of future research utilizing biomarkers such as SmO2 levels and arm torque to monitor performance and training load, and to develop effective exercise and recovery protocols for athletes. Evaluating these biomarkers may offer valuable insights for designing studies that contribute to the advancement of wearable sensor technology research and its application in monitoring physiological health. These trends provide a foundation for future studies to adopt similar designs or formulate relevant hypotheses, thereby enhancing the credibility of wearable sensor technologies in orthopedics. Additionally, constructing computational models to simulate the kinematics of extremities based on a comparison of biological and theoretical data can facilitate predictions of athletes’ training tolerances and performance using the physiological data obtained from these sensors.

Biomarker trends shed light on how athletes physiologically respond to overarm movements. Similarly, these biomarkers can help assess stress on other ligaments that are commonly affected in sports injuries, like Anterior Cruciate Ligament (ACL) or Medial Collateral Ligament (MCL) tears. The information gleaned from SmO2 levels provides insights into the kinetic chain of various movement patterns in sports, such as quick cuts, hip rotations, and lateral shuffles. Therefore, the real-time monitoring of these biomarkers is crucial for making immediate adjustments and optimizing performance. Continuous monitoring during training enables assessment of the athlete’s physiological response, aiding in preventing overexertion and reducing the risk of overuse injuries.

The current field of rehabilitation and return-to-play protocols lacks continuous objective measurements to assist physicians and physical therapists in ligament health assessments. For example, for ulnar collateral ligament injuries, current rehabilitation processes utilize objective testing to determine the athlete’s readiness for return to sport, specifically assessing the tolerance of athlete’s musculoskeletal function and movement [36,37]. Therefore, standardized return-to-play protocols incorporating an assessment of the biomarkers in accompaniment with physical trials could serve as an additional measure to assess an athlete’s return to play. Rehabilitation protocols could be developed around the redevelopment of objective means derived from the return-to-play protocol.

## Figures and Tables

**Figure 1 sensors-24-04710-f001:**
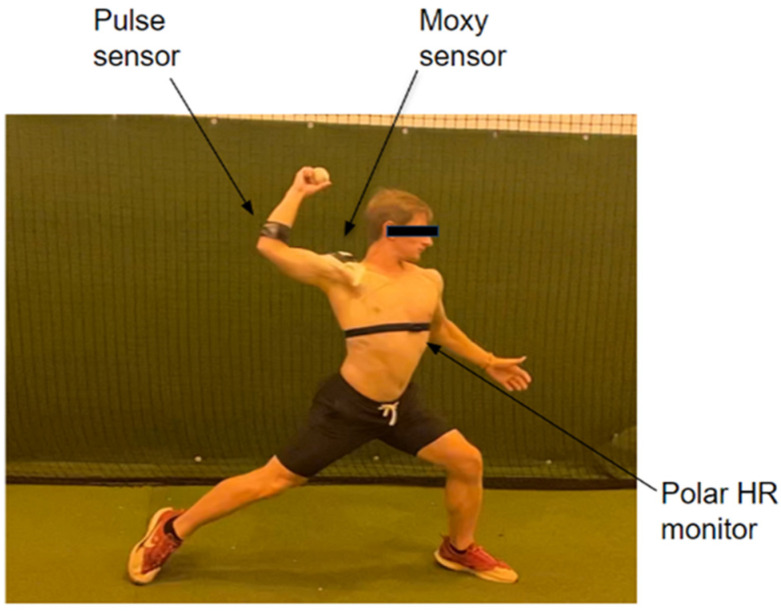
A schematic representing the placement of the Moxy, Driveline Pulse sensor, and HR monitor for each subject in the trial.

**Figure 2 sensors-24-04710-f002:**
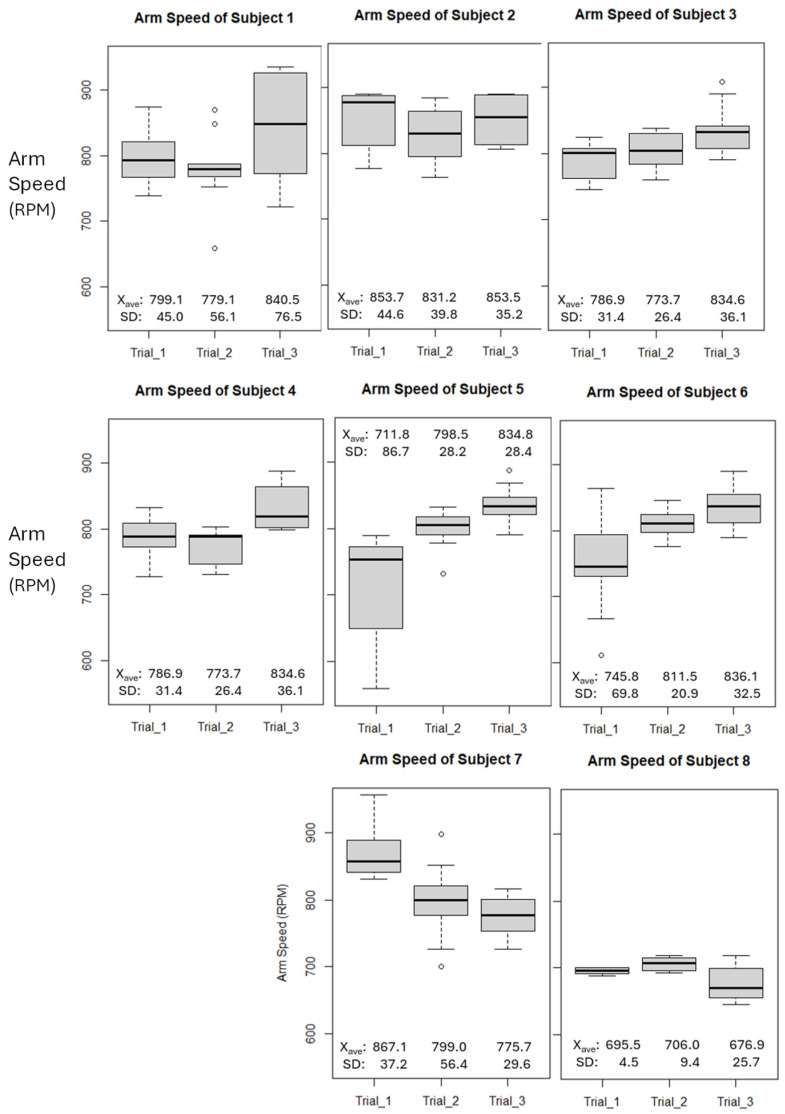
Box and Whisker plots of arm speed, measured by Pulse sensor for each subject. Each plot contains arm speed values for each trial, containing 10 pitches.

**Figure 3 sensors-24-04710-f003:**
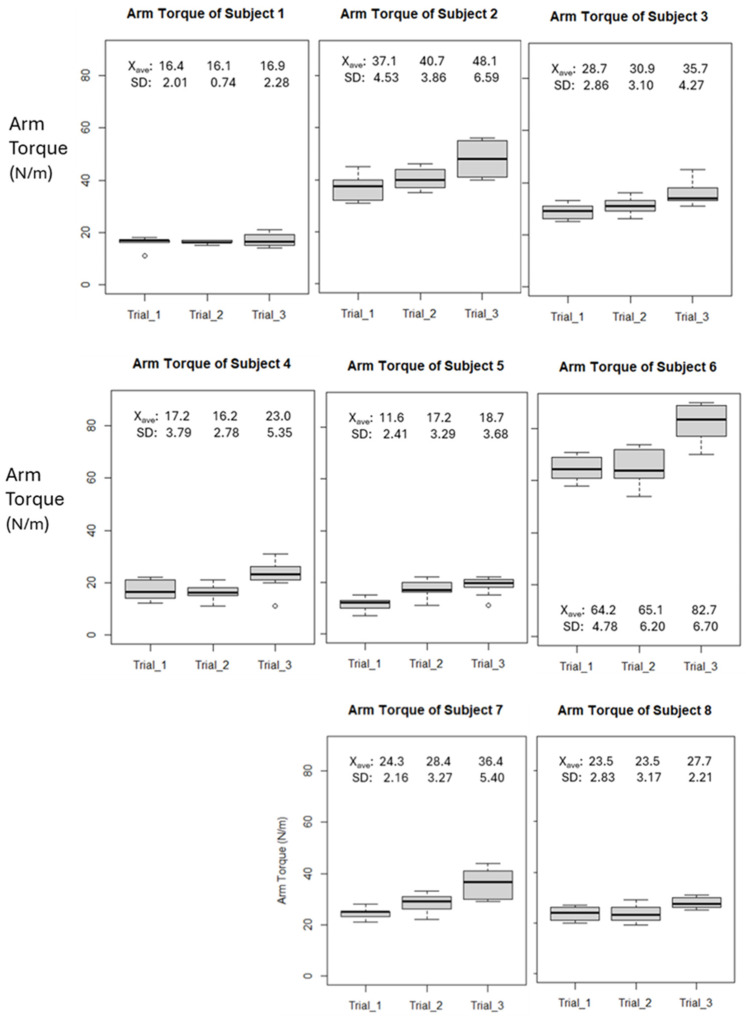
Box and Whisker plots of arm torque, measured by Pulse sensor for each subject. Each plot contains arm torque values for each trial, containing 10 pitches.

**Figure 4 sensors-24-04710-f004:**
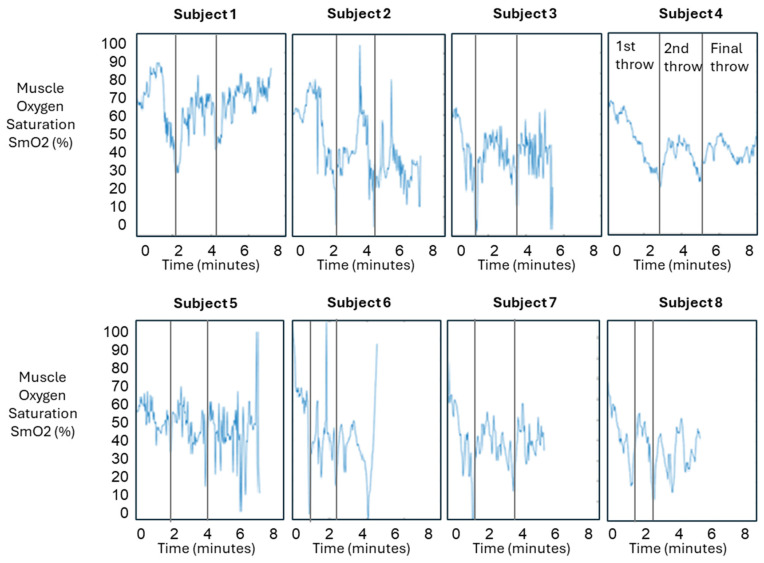
Plots of muscle oxygen saturation (%) measured by Moxy sensor. Each plot contains muscle oxygen saturation values for each participant in the trial. The data shown concatenate the SmO2 data for all three trials shown in Table 1. Each plot is divided into three ranges corresponding to the first, second, and last of the ten throws.

**Figure 5 sensors-24-04710-f005:**
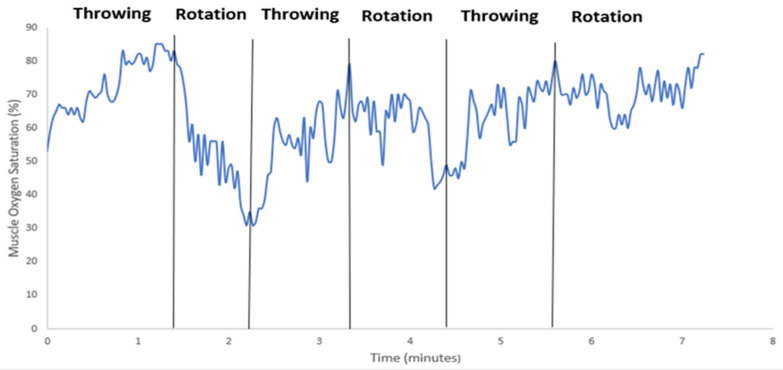
Continuous Sm02 tracing produced by Moxy sensor for Participant 1 during the baseball trial for a period of eight minutes.

**Figure 6 sensors-24-04710-f006:**
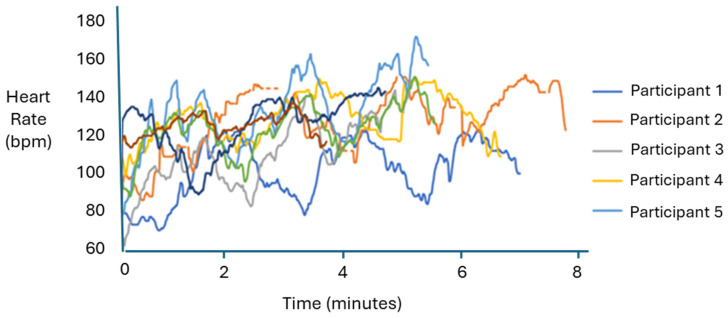
A plot representing heart rate (bpm) data for all eight participants over an eight-minute trial period. These combined heart rate data are not intended to identify specific trends but to illustrate the safe range of heart rate values experienced by participants during the trial. The chart provides an overview of the variability and safety margins in heart rate responses under the given exercise conditions.

**Table 1 sensors-24-04710-t001:** Representation of the protocol placed in the pilot study for eight healthy collegiate baseball athletes from Case Western Reserve University, who participated in the study across the 14-week season.

Exercise	Description
A. Warm-up	Collegiate team’s warm-up.
B. Throws	10 throws at “game speed”.
C. Deltoid circuit	Using either a Drive Line resistance band or 5:lb weighted disc: Elevated internal rotation ×8; Front deltoid raise ×8; Elevated external rotation ×8.
D. Throws	10 throws at “game speed”.
E. Deltoid circuit	Using either a Drive Line resistance band or 5:lb weighted disc: Elevated internal rotation ×8; Front deltoid raise ×8; Elevated external rotation ×8.
F. Throws	10 throws at “game speed”.
G. Deltoid circuit	Using either a Drive Line resistance band or 5:lb weighted disc: Elevated internal rotation ×8; Front deltoid raise ×8; Elevated external rotation ×8.

**Table 2 sensors-24-04710-t002:** Representation of the eight healthy collegiate baseball athletes from Case Western Reserve University who participated in the study across the 14-week season, including their position, weight, height, and year.

Participant	Position	Weight	Height	Year
Subject 1	Center	185 lb	5′11″	2
Subject 2	Left-handed pitcher	190 lb	5′11″	3
Subject 3	Left-handed pitcher	185 lb	6′0″	2
Subject 4	Right-handed pitcher	187 lb	5′10″	4
Subject 5	Right-handed pitcher	200 lb	6′5″	3
Subject 6	Center	195 lb	6′0″	1
Subject 7	Right-handed pitcher	190 lb	5′9″	3
Subject 8	Outfield	205 lb	6′3″	2

**Table 3 sensors-24-04710-t003:** Trend line approximations from (a) the slope of a linear regression for just the mean values of Trial 1, Trial 2, and Trial 3; (b) the percentage change in value of the measured torque between the endpoint (mean of Trial 3) and the initial value (mean of Trial 1).

Subject	Player Position	m, from Regression	End-Point % Change
1	Center	0.3	3.1
2	Pitcher	5.5	29.7
3	Pitcher	3.5	24.4
4	Pitcher	2.9	33.7
5	Pitcher	3.5	61.2
6	Center	9.0	27.8
7	Pitcher	6.1	49.8
8	Outfield	2.1	17.9
	Mean=	4.1	30.9

## Data Availability

The raw data supporting the conclusions of this article will be made available by the authors on request.

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
