# Peer review of "Overarm Training Tolerance: A Pilot Study on the Use of Muscle Oxygen Saturation as a Biomarker"

_sensors, 2024, doi:10.3390/s24144710_

Round 1

Reviewer 1 Report

Comments and Suggestions for Authors

In this manuscript, a study of muscle Oxygen saturation as a biomarker is presented. However, there are some problems in the manuscript.

1.      Please check the manuscript for format.

2.      The experiments and the results are vague and difficult to understand. Experiments and results should be described in detail. And the results must be well analyzed.

I suggest to reject this manuscript.

Comments on the Quality of English Language

Minor editing of English language required

Author Response

The senior author of the manuscript appreciates these candid comments.  The writing of the manuscript was an opportunity for student researchers to lead the narrative and it in this case the lead author failed to adequately review the drafts.  The paper had been substantially re-written with clearer explanations and improved grammar and essay structure. Journal formatting instructions were consulted and many formatting changes made.

We believe the research conducted and results are worthy of dissemination, and hope the revised manuscript has a more acceptable quality.

Reviewer 2 Report

Comments and Suggestions for Authors

This manuscript reports on the the sport monitoring using wearable sensors. The athlete participators are all normal and the results show the correlation between the torque, speed, oxygen saturation and the rate of the participators after throwing games. The trend can be seen mostly in the figures, but the results seem not so directly correlative to the injury potential, particularly for sports, since the sports motion process is complicated and with variation among individual tests. Considering the results and contribution, I recommend the manuscript be accepted after addressing some questions as below.

1. More details about the experimental set-up should be enclosed in the manuscipt, including the decription of the wearable sensors, the data or analysis for the correlation between the indication of varied tests.

2. What are the novelty for using these sensors to monitor the performance of  atheletes? 

3. Please also compare the results of different sensors as well as the advantages and disadvantages.

4. The figures are not very clear for demonstration, and please make some improvement.

Comments on the Quality of English Language

The quality of this manuscript is quite good.

Author Response

1. More details about the experimental set-up should be enclosed in the manuscipt, including the decription of the wearable sensors, the data or analysis for the correlation between the indication of varied tests.

The section on experimental set-up has been extensively expanded, and the choice of sensors is more clearly justified, with references.  We did not repeat trials using a different sensor suite, and thus have limited comments on sensor comparisons – we essentially relied on the literature and our prior work.

2. What are the novelty for using these sensors to monitor the performance of atheletes?

3. Please also compare the results of different sensors as well as the advantages and disadvantages.

Combined response to C2 and C3. The novelty of our sensor suite is also addressed in the revised manuscript, lines 134-158.  We present that the specific sensor suite has predicate use in other baseball and overarm studies.

4. The figures are not very clear for demonstration, and please make some improvement.

A significant amount of time was spent re-plotting and reformatting figures for clarity so as to Journal quality.  We believe the larger and less cluttered look of the figures represents a significant improvement.

Reviewer 3 Report

Comments and Suggestions for Authors

Thank you so much for the review invitation for "Overarm Training Tolerance: A Pilot study on Muscle Oxygen 2 Saturation as a Biomarker" by Gorti et al.. Here are the major and minor comments: 

Major comments:

  • The results section includes multiple plots, but an explanation of the results needs to be added. Please consider adding some interpretations and explanations for the results/observations. For lines 230-232 and 243 -251, the results seem more fitting. 
  • Please provide more information on which trial and when the muscle oxygen saturation (%) was plotted. The lines 230 and 232 show different ranges; were these from these plots? 

Minor comments:

  • Extract period on line 77. 
  • In Figure 2, 3 and 4. Please consider increasing the resolution of the image. Also, please clearly state that the red line is the median or mean and the outlier in the red plus symbol, right? Also, there is a gap on some of the bars; why was there a break? Also, for subject 7 in Figure 2, Trial 1, the Q3 blue bar is missing. Was there a printing error? Subject 7 Trial 1 in Figure 3 is out of bounds; please consider either increasing the y limits or notating the upper limits in the legend. 
  • Lines 201-203 should be in the Methods. 
  • Figure 4 Subject 1 is masked by some symbols, please update the plot. 

Author Response

Major comments:

The results section includes multiple plots, but an explanation of the results needs to be added. Please consider adding some interpretations and explanations for the results/observations. For lines 230-232 and 243 -251, the results seem more fitting.

The manuscript has been extensively re-written and the results section – previously void of meaningful content –now has more extensive narrative and interpretations to accompany the results.  As mentioned to Reviewer 1, the senior author of the manuscript appreciates the requests for clarification; the original writing of the manuscript was an opportunity for student researchers to lead the narrative and it in this case the lead author failed to adequately review the drafts.  This has been corrected.

Please provide more information on which trial and when the muscle oxygen saturation (%) was plotted. The lines 230 and 232 show different ranges; were these from these plots?

The details you requested are an oversight in production. Due ot the way the Moxy works, we did a start/stop for each of the throwing and rotation exercises. Each plot contains muscle oxygen saturation values for each participant in the trial.  The data shown concatenates the SmO2 data for all three trials shown in Table 1.

Minor comments:

Extract period on line 77.

Corrected.

In Figure 2, 3 and 4. Please consider increasing the resolution of the image. Also, please clearly state that the red line is the median or mean and the outlier in the red plus symbol, right? Also, there is a gap on some of the bars; why was there a break? Also, for subject 7 in Figure 2, Trial 1, the Q3 blue bar is missing. Was there a printing error? Subject 7 Trial 1 in Figure 3 is out of bounds; please consider either increasing the y limits or notating the upper limits in the legend.

All images have been recreated with what we believe is a cleaner, clearer, presentation. This corrected the issue that have been presented.

Lines 201-203 should be in the Methods.

Lines 201 and 203 noted as incorrectly placed have now been included in the Methods section.

Figure 4 Subject 1 is masked by some symbols, please update the plot.

Plotting errors have been corrected by using a more contemporary plotting system (R vs MATLAB)

Round 2

Reviewer 1 Report

Comments and Suggestions for Authors

In this manuscript, a study of muscle Oxygen saturation as a biomarker is presented. However, there are some problems in the manuscript.

1.      Please check the manuscript for format.

2.      The experiments and the results are vague and difficult to understand

I suggest to reject this manuscript.

Comments on the Quality of English Language

Minor editing of English language required

Reviewer 2 Report

Comments and Suggestions for Authors

The authors have addressed most of my questions. This manuscript can be accepted for publication.

Author Response

Thank you for your comments on the revised manuscript.  We have revised the manscript again and feel that we now meet the standard for improve clarity of result presentaion.

Reviewer 3 Report

Comments and Suggestions for Authors

Thank you so much for the revision invitation for "Overarm Training Tolerance: A Pilot study on Muscle Oxygen 2 Saturation as a Biomarker" by Gorti et al.. The authors address my comments, and here are the major and minor comments with suggestions based on the new information added to the revised manuscript: 

Major comments:

  • It would be better to add a guidance range in Figure 4 to highlight the first, second, and final of the ten throws, as the Time it took each subject varies, and did these include the Deltoid circuit period? Similar to what you have done for Figure 5. Given there are only eight subjects, a plot with eight rows will definitely help. 

Minor comments:

  •  The linear regression used on line 280 and later mention of the "trend lines" should be detailed in the Methods section. 
  •  Would it be possible to fix the Time (minutes) in Figure 6 to an integer, as 0:02:53 seems a bit odd? Also, please adjust the Excel plot area to make sure the legend displays the full Participant # text for better visuals. 
  • I do not think repeating the focus of the study verbose is necessary lines 463-467 and 168-171. 

Suggestion:

  • From a statistical point of view, instead of running the different models and obtaining different m and % endpoint changes, could a linear mixed effects model work better? Essentially, the fixed effect will be the player position, and the random effect will be the subjects; this way, the results will be more generalizable as the results are done using multiple subjects (aimed to represent the population better). Just a suggestion: I doubt this would change your conclusions. 
